# Comprehensive Analysis of *SnRK* Gene Family and their Responses to Salt Stress in *Eucalyptus grandis*

**DOI:** 10.3390/ijms20112786

**Published:** 2019-06-06

**Authors:** Yujiao Wang, Huifang Yan, Zhenfei Qiu, Bing Hu, Bingshan Zeng, Chonglu Zhong, Chunjie Fan

**Affiliations:** 1Key Laboratory of State Forestry Administration on Tropical Forest Research, Research Institute of Tropical Forestry, Chinese Academy of Forestry, Guangzhou 510520, China; wangyujiao0207@163.com (Y.W.); huifangy@sina.com (H.Y.); qzf5720@ritf.ac.cn (Z.Q.); hubinghello@163.com (B.H.); b.s.zeng@vip.tom.com (B.Z.); zclritf@126.com (C.Z.); 2Key Laboratory of Forest Genetics and Biotechnology, Ministry of Education of China, Co-Innovation Center for the Sustainable Forestry in Southern China, Nanjing Forestry University, Nanjing 210037, China

**Keywords:** *Eucalyptus grandis*, SnRK family, bioinformatics analysis, salt stress, *Cis*-elements

## Abstract

The sucrose non-fermentation-related protein kinase (SnRK) is a kind of Ser/Thr protein kinase, which plays a crucial role in plant stress response by phosphorylating the target protein to regulate the interconnection of various signaling pathways. However, little is known about the SnRK family in *Eucalyptus grandis*. Thirty-four putative SnRK sequences were identified in *E. grandis* and divided into three subgroups (SnRK1, SnRK2 and SnRK3) based on phylogenetic analysis and the type of domain. Chromosome localization showed that SnRK family members are unevenly distributed in the remaining 10 chromosomes, with the notable exception of chromosome 11. Gene structure analysis reveal that 10 of the 24 *SnRK3* genes contained no introns. Moreover, conserved motif analyses showed that SnRK sequences belonged to the same subgroup that contained the same motif type of motif. The Ka/Ks ratio of 17 paralogues suggested that the *EgrSnRK* gene family underwent a purifying selection. The upstream region of *EgrSnRK* genes enriched with different type and numbers of cis-elements indicated that *EgrSnRK* genes are likely to play a role in the response to diverse stresses. Quantitative real-time PCR showed that the majority of the *SnRK* genes were induced by salt treatment. Genome-wide analyses and expression pattern analyses provided further understanding on the function of the SnRK family in the stress response to different environmental salt concentrations.

## 1. Introduction

Plants are confronted with various environmental insults in nature, such as drought, salt, low temperature and pathogen attack. Under these situations, plants are capable of initiating their own defense mechanism to adapt to various potentially stressful damaging challenges. The process of phosphorylation and dephosphorylation of proteins represents very important mechanisms for plants to respond to environmental stress signals [1]. Protein kinase is an important regulator in plants, which senses environmental signals through membrane receptor proteins and activates different protein phosphorylation pathways, which regulate the expression of downstream stress-resistant genes, and protect plants or reduce harm to those plants from an adverse external environment [2]. In recent years, various studies have reported on protein kinase genes that are related to resistance, namely the receptor protein kinase (RLK) [3], mitogen-activated protein kinase (MAPK) [4], calcium dependent protein kinase (CDPK) [5], and sucrose nonfermenting1 (SNF1) kinase families [6].

SnRK is a Ser/Thr protein kinase, which widely exists in plants, including SnRK1, SnRK2 and SnRK3 subfamilies [6,7]. SnRK1 is homologous to yeast and mammals and has a highly conserved terminal catalytic domain, which plays an important role in regulating carbon metabolism and energy status [8]. The SnRK2 and SnRK3 subfamilies are unique to plants, both of which have additional members and are more diverse than their SnRK1 subfamily counterparts. In addition to a highly conserved kinase domain at the N-terminus, SnRK2s also contains a variable adjusting conserved domain at the C-terminus [9]. SnRK3 can bind to calcineurin B-like (CBL) protein to participate in the Ca^2+^-mediated stress response, and is referred to as the calcineurin B-like protein-interacting protein kinases (CIPK). The N-terminal of CIPK has a conserved kinase domain and the C-terminal has two conserved domains: NAF and PPI [10,11].

The SnRK1 subfamily is primarily involved in metabolic regulation and affects plant development. In addition, the SnRK1 gene has been isolated from many plants [7]. The SnRK1 gene family is a relatively small subfamily. In *Arabidopsis thaliana*, the SnRK1 gene family contains *AtSnRK1.1*, *AtSnRK1.2* and *AtSnRK1.3* [12]. In transgenic barley, expression of antisense SnRK1 protein kinase affects pollen development, resulting in infertility [13]. Furthermore, a decrease of SnRK1 in pea activity by 50 - 70 percent will lead to the increased of sucrose accumulation and the defective of seed maturation [14]. In addition, SnRK1 is similar to yeast SNF-1, which is involved in carbon metabolism [8]. Substantial evidence suggests that SnRK1 is involved in the regulation of various physiological and biochemical processes in plants, and it has been shown to connect stress and metabolism [15].

SnRK2 plays an important role in the ABA signal transduction pathway, osmotic stress and sugar metabolism. In *A. thaliana*, eight out of ten *SnRK2* genes can be induced by hyperosmotic and salt stress, and five are activated by ABA [16]. Specifically, the stomatal closure and ABA mediated gene expression in the process of *AtSnRK2.6* involved in ABA regulation [17,18]. Furthermore, an overexpression of *AtSnRK2.8* shows higher resistance to drought [19]. At present, this subfamily has been extensively studied in different plants, such as rice [20], corn [21], soybean [22], cotton [23], *Brachypodium distachyon* [24], and *Hevea brasiliensis* [25]. All SnRK2 members are induced by one or more abiotic stresses, especially salt and osmotic stress, and includes overexpression of *SAPK* and *NtSnRK2.1* enhanced salt tolerance of transgenic plants [26,27]. *PtSnRK2.5* and *PtSnRK2.7* positively regulate plant responses to salt stress [28]. The overexpression of *TaSnRK2.3*, *TaSnRK2.4*, *TaSnRK2.7* and *TaSnRK2.8* in *A. thaliana* was reported to enhance plants tolerance to salt and other stresses [29,30]. Recent studies have shown that *TaSnRK2.9* improved salt and drought stress tolerance through detoxification of (reactive oxygen species) ROS [30]. Additionally, overexpression of *GhCIPK6* significantly increased the tolerance of transgenic *A.thaliana* to salt, drought and ABA stress [31]. The *ZmSnRK2.8* has a positive regulating effect on the salt stress signal transduction pathway, while the *ZmSnRK2.11* is a possible negative regulator in response to the salt and drought stress signal transduction pathways [32]. These findings suggest that members of the SnRK2 subfamily play an important role in salt and other abiotic stresses.

The SnRK3 subfamily also responds to abiotic stress. For example, *OsCIPK31* was sensitive to salt and other stressful conditions during germination and seedling growth in rice [33]. *A. thaliana CIPK6* was involved in growth and development. Furthermore, *atcipk6* reduces auxin transport and tolerance to salt stress [34]. Overexpression of *SICIPK24* (*SISOS2*) enhanced salt tolerance in the tomato [35]. The expression of *AtCIPK3* increased the tolerance of *A.thaliana* to various stress stimuli, including high salt concentrations, wounds, drought, etc. [36]. In addition, *AtCIPK21* regulates osmotic and salt stress responses [37]. Further, SnRK3s can interact with CBL, and the CBL-CIPKs complex is a complicated calcium-signaling system that performs a vital function in resistance to various stresses in plants [6,38]. One of the most famous mechanisms of the SOS (salt overly sensitive) system is provided by *SOS2/AtCIPK24* (salt overly sensitive 2), which is a member of the SnRK3 subfamily in *A.thaliana*. As Na^+^/H^+^ antiporter, it improves plant salt tolerance by maintaining ionic homeostasis [39,40]. The overexpression of *PtSOS2* enhances salt tolerance by mediating osmotic protection and inducing the protective antioxidant enzyme system [41]. Collectively, the above-mentioned studies demonstrated that *CIPKs* play important roles in the physiological response to salt tolerance in different plants.

Although the SnRK gene family plays an important role in the abiotic stress response, and the ABA signaling pathway and development, there is still a sustained lack of research on understanding the functional import of the *SnRK* gene in *Eucalyptus grandis. E*. *grandis* has tremendous economic, ecological and social value. As an excellent timber forest, it provides a large number of raw materials for both the paper industry and timber industry. Today, it occupies a huge cultivation area. With the rapid development of molecular biology and the publication of the *E. grandis* genome [42], it is possible for us to analyze the *SnKR* gene family at the genome-wide level. In this study, the *SnRK* gene family was systematically studied in *E. grandis* using bioinformatics. A total of 34 *SnRK* genes were identified. In addition, their phylogenetic relationships, gene structures, protein motifs, chromosomal location and promoter were analyzed. Further, the expression patterns of the *SnRK* gene family in different tissues were calculated while the differential expression of the *SnRK* gene was executed under different salt concentrations and salt treatment duration by qRT-PCR. These results will lay the foundation for further study of the molecular mechanisms that account for salt tolerance and molecular breeding in *E. grandis*.

## 2. Results

### 2.1. Identification of SnRK Genes in E. grandis

A total of 34 candidate genes were identified in the *E. grandis* genome. Based on the subfamily and their physical location (from top to bottom) on the chromosome, *EgrSnRK* genes were named *EgrSnRK1.1*~*EgrSnRK1.2*, *EgrSnRK2.1*~*EgrSnRK2.8*, and *EgrSnRK3*.1~*EgrSnRK3.24*. The parameters of the gene characteristics including the open reading frame (ORF) length, chromosome location, exons, protein molecular weight (MW) and isoelectric point (pI) were analyzed and are shown in Table 1. They encode proteins that range from 334 to 550 amino acids (aa) in size, with an average of 427 aa, with a molecular weight that varies from 38.07 kDa (*EgrSnRK2.2*) to 61.62 kDa (*EgrSnRK3.11*). Moreover, the theoretical isoelectric point (pI) ranged from 4.73 to 9.52; however, the isoelectric points of the SnRK2 subfamily were distributed from 4.73 to 6.18, and these belong to acidic proteins. The predicted subcellular localization data (Appendix A) showed that 82.4 percent of the *EgrSnRK* genes were predicted to be expressed in the nucleus and cytoskeleton, followed by 73.5 percent in the chloroplast. Additionally, gene expression of only *EgrSnRK3.9*, *EgrSnRK3.11* and *EgrSnRK3.14* genes was predicted in peroxisomes. Relevant details are shown in Appendix A.

### 2.2. Phylogenetic Tree of SnRK Genes

For the purpose of researching the evolutionary relationships of *SnRK* genes in *A. thaliana*, rice, grape, poplar and *E. grandis*, a phylogenetic tree was built with 34, 39, 48, 30 and 45 SnRK protein sequences, respectively, which was constructed using MEGA 7.0 by employing the Neighbor-Joining (NJ) methods with 1000 bootstrap replicates (Appendix A). The accession numbers or locus IDs of the *SnRK* genes are listed in Table 1 and Appendix A. The phylogenetic analysis (Figure 1) indicated that the *SnRK* genes could be divided into three groups in combination with an analysis of the Ser/Thr Kinase domain by Pfam, NCBI and SMART. The SnRK3 subfamily has the largest number of members, while the SnRK1 subfamily has the fewest members and includes two to four genes. In addition, the SnRK2 subfamily has eight members. The *SnRK* genes of each group were evenly distributed, but it was found that the clustering distribution of *SnRK* genes in rice was obvious, which might be due to the fact that rice is a monocotyledonous plant and the other species are dicotyledonous plants.

### 2.3. Multiple Sequence Alignment of the SnRK Gene Family

In order to further explore the structural features of the EgrSnRK family, multiple sequence alignment was performed with 32 amino acid sequences by DNAMAN 8. Results showed that all SnRKs genes could be divided into three subfamilies, in which the EgrSnRK2 subfamily included highly conserved domains at the N- and C terminals, which had divergent domains (Figure 2). All the members of the EgrSnRK2 subfamily had an ATP binding site and the serine/threonine protein kinase active-site in the kinase domains of their N-terminal regions. Moreover, there are two different domains at the C-terminal, of which domain I was necessary for osmotic stress-mediated activation and exists in all members of the EgrSnRK2 subfamily. Domain II was only present in strongly ABA-responsive kinases. Domain II was also observed only in *EgrSnRK2.6* and *EgrSnRK2.7*. Similarly, the SnRK3 subfamily had a protein kinase domain at the N-terminal, while a NAF region was observed at the C-terminal. NAF domains played an important role in interacting with CBLs.

### 2.4. Motif Composition and Gene Structural Analysis of the SnRK Gene Family in E. grandis

To reveal the intron/exon structure of EgrSnRK family genes, we combined the online Gene Structure Display Server and phylogenetic tree for subsequent analysis (Figure 3). The result showed that members of the same subfamily were similar. The SnRK1 subfamily (*EgrSnRK1.1* and *EgrSnRK1.2*) have 10 exons, and all SnRK2 genes have nine exons, which was similar to cotton, *Hevea brasiliensis* and *Malus prunifolia* [25,43]. However, the number of exons present in the SnRK3 subfamily showed obvious differences, which varied from 1 to 15. Furthermore, there were five genes that contained one exon and one gene contained two exons, the remainder of the 18 SnRK3 genes contained multiple extrons ranging from 5 to 15.

The sequence of the EgrSnRK protein was further analyzed using MEME. Twenty different conserved motifs were acquired from MEME and sequence and length information of the conserved motif are shown in Appendix A. Each of the putative motifs obtained from MEME was annotated by searching Pfam. Motif 1 and motif 2 encoded a protein kinase domain, while motif 10 and motif 11 encode a NAF domain. As for the other motifs, these do not have a functional annotation. It was interesting to note that all *EgrSnRK* genes included the same conserved motifs (motif 1/2/3/4/5/7/9/14; Figure 4). Meanwhile, we found that the SnRK1 and SnRK2 subfamily share a common type and position of the conserved motif. Besides, Motif 19 is specific to the SnRK1 subfamily, while Motif 12 and 17 only appear in the SnRK2 subfamily and other motifs were found in the SnRK3 subfamily.

### 2.5. Chromosomal Location and Gene Pairs Analysis in E. grandis

A chromosome localization map was constructed with the location information of *EgrSnRK* genes. The results showed that 34 *EgrSnRK* genes were distributed unevenly on 10 chromosomes (Figure 5), and chromosome 11 contained no genes that belonged to the SnRK family. Moreover, only one *EgrSnRK* (*EgrSnRK3.19*) gene was distributed on chromosome 6, while the most *SnRK* genes (6 genes) were distributed on chromosome 1. We found that the SnRK3 subfamily genes were mainly distributed on chromosome 1, 2, 3 and 10. In addition, clusters are formed on chromosomes 1, 2 and 7. Besides, one gene cluster, namely *EgrSnRK3.1/EgrSnRK3.2/EgrSnRK3.3/EgrSnRK 3.4/EgrSnRK3.5*, appeared to have evolved from tandem duplication events, which *EgrSnRK3.1/EgrSnRK3.2* and *EgrSnRK 3.4/EgrSnRK3.5* by tandem duplication (box in Figure 5) according to previous studies [42]. Due to the different types and functions of genes, the rate of evolution varies. Thus, to explore the role of selection pressure in *SnRK* gene family evolution, Ks values, Ka values, and Ka/Ks ratios of paralogues and orthologues were obtained and from this we built a sliding-window analysis for paralogues genes. Details of paralogues and orthologues are listed in Table 2 and Appendix A. In fact, a Ka/Ks = 1 represents neutral selection, a Ka/Ks>1 shows positive selection and a Ka/Ks>1 is indicative of purifying selection. The Ka/Ks ratio of 17 paralogous gene pairs was less than 1, with the exception of the *EgrSnRK3.1/EgrSnRK3.4* (1.035). The result suggested that the *EgrSnRK* gene family underwent purifying selection.

### 2.6. Promoter Analysis

To better understand the function and regulation of *EgrSnRK* genes, the promoter sequences of all genes were analyzed by using PlantCARE. A series of cis-elements related to the abiotic stress response, plant hormone response and plant growth and development were identified (Appendix A). There are light responsiveness (G-Box, GT1-motif and I-box, etc.) and hormone-responsive elements, including the gibberellin-responsive, MeJA-responsiveness, ethylene-responsive, salicylic acid responsiveness, abscisic acid responsiveness and auxin-responsive, abiotic stresses elements (LTR, MBS and WUN-motif). In addition, some elements are involved in plant growth and development (i.e., CAT-box, HD-Zip 1, GC-motif, etc.). Five categories of cis-acting elements involved in the abscisic acid responsiveness, low-temperature responsiveness, drought-inducibility, wound-responsive and defense and stress responsiveness are shown in Figure 6. In addition, the number of these cis-acting elements in each *EgrSnRK* gene is shown in the Table 3. It was also found that *EgrSnRK* genes contained a variety of cis-acting elements of different types and the same element displayed multiple sites in the promoter region of one *EgrSnRK* gene.

### 2.7. EgrSnRK Genes Expression under NaCl Treatment

To investigate whether the *EgrSnRK2* and *EgrSnRK3* genes were responsive to salt stress, the expression of *EgrSnRK2s* and *EgrSnRK3s* in different tissues (Root, Stem and leaf) of the *E. grandis* were detected by qRT-PCR after different concentrations of NaCl were used to treat eucalyptus seedlings for different time point. In addition, the expression of *EgrSnRK3.1* was not detected in this experiment.

Figure 7 showed the expression pattern of different concentrations under conditions of a 200mM salt treatment. The expression levels of some genes (i.e., *EgrSnRK3.10*, *EgrSnRK3.12*, *EgrSnRK3.18*, *EgrSnRK3.21* and *EgrSnRK3.22*) decreased to less than half that of CK (untreated seedlings) at different time points. It is clear that the expression levels of the *EgrSnRK* genes did not change significantly or were inhibited in the stem, while the expression of *EgrSnRK2.1*, *EgrSnRK2.3*, *EgrSnRK3.9* and *EgrSnRK3.13* were up-regulated more than 4-fold. Among the 31 genes in the leaf tissue, 12 of them were up-regulated, among which *EgrSnRK2.1* and *EgrSnRK2.2*, *EgrSnRK3.3*, *EgrSnRK3.4*, *EgrSnRK3.9* and *EgrSnRK3.13* were up-regulated by more than 10-fold higher at certain concentrations. Specifically, the expression of the *EgrSnRK3.13* was up-regulated more than 5-fold at low concentrations but reached high expression (more than 20-fold) levels when exposed to a much higher 200-400mM concentration. Additionally, some genes (i.e., *EgrSnRK2.4*, *EgrSnRK3.2*, *EgrSnRK3.7*, *EgrSnRK3.8* and *EgrSnRK3.9*, *EgrSnRK3.14* and *EgrSnRK3.23*) were up-regulated at its maximum value after 400 mM treatment, in which both *EgrSnRK3.7* and *EgrSnRK3.8* were inhibited at low concentrations and up-regulated at the maximum concentration.

To further investigate whether or not *EgrSnRKs* were responsive to salt stress, different time periods of salt treatment were used. As shown in Figure 8, Of the 31 *EgrSnRK* genes, 12 of them (i.e., *EgrSnRK2.1*, *EgrSnRK2.3*, *EgrSnRK2.8*, *EgrSnRK3.4*, *EgrSnRK3.6*, *EgrSnRK3.8*, *EgrSnRK3.11*, *EgrSnRK3.13*, *EgrSnRK3.14*, *EgrSnRK3.15*, *EgrSnRK3.17* and *EgrSnRK3.24*) were up-regulated at different time points, while only the expression of the *EgrSnRK3.10* gene was down-regulated at all time points in different tissues (root, stem and leaf; Figure 7). Interestingly, the expression of the *EgrSnRK2.1*, *EgrSnRK3.13* and *EgrSnRK3.10* genes showed similar expression patterns in different tissues. For example, the expression levels of *EgrSnRK3.13* increased over time. Some Eucalyptus genes are induced to be expressed more strongly than others. For example, the expression of *EgrSnRK2.1*, *EgrSnRK2.2*, *EgrSnRK3.2* and *EgrSnRK3.3* in the root tissue and expression of *EgrSnRK2.1*, *EgrSnRK2.4*, and *EgrSnRK3.2* in the stem tissue were all up-regulated by a factor of more than five-fold. Of the 31*EgrSnRK* genes, 17 genes in the root, 29 genes in the leaf and 11 genes in the stem were up-regulated, respectively.

## 3. Discussion

The SnRK family plays an important role in the response to stress. Hence, the SnRK family and subfamily were analyzed in genome-wide studies in many plants, including *A. thaliana*, rice, distachyon, cotton, *Hevea brasiliensis*, and *Vitis vinifera*. However, the SnRK gene family has not been identified in *E. grandis*.

In this study, we identified 34 SnRK family members including two SnRK1 genes, eight SnRK2 genes and 24 SnRK3 (CIPK) genes in *E. grandis*. The phylogenetic analysis showed that similar members of the SnRK1 and SnRK2 subfamilies in diverse species, while the members of the SnRK3 subfamilies were the lowest (20) in the grape and the highest in rice (34). When compared with other species, the number of *EgrSnRK* genes in each subfamily was similar to the grape. Furthermore, many *SnRK* genes are clustered on the terminal branches of the phylogenetic tree, and the sequence similarity between some gene pairs was very high. Previous studies revealed that different subfamilies of the *SnRK* gene family had different conserved domains, but all genes had a protein kinase domain at the N-terminus. The gene sequences of genes belonging to the same subfamily were highly conserved and shared the same type of domains, indicating the functional diversity of the SnRK gene family (Figure 2).

Exon-intron structural diversification and motif composition played an important role in the evolution and function of many gene families. The number of exons varied in different subfamilies, and genes; the *EgrSnRK1.1* and *EgrSnRK1.2* genes had 10 exons, which was the same as that reported for *BdSnRK1s* [24]. Furthermore, *EgrSnRK2s* had nine exons, which was the same number as reported for *VvSnRK2s* and most *AtSnRK2s,* with the exception of *AtSnRK2.6*(11), *AtSnRK2.4* (8) and *AtSnRK2.8* (6) [44]. For the SnRK3 subfamily, in each reported species, we showed that there were either members with more than 10 exons or members with only one. The Figure 3 shows that most paralogous gene pairs contain the same exon number, although some gene pairs have different exon numbers. For instance, *EgrSnRK3.14* and *EgrSnRK3.20* lost one intron as compared with *EgrSnRK3.11* and *EgrSnRK3.21*. To investigate conserved motifs in more detail, we determined the number and type of conserved motif for all *EgrSnRK* genes. The results indicated that the types and numbers of the motifs in the same subgroup were the same (*EgrSnRK*1 and *EgrSnRK2;*
Figure 4), an observation that suggested a close evolutionary relationship within the subgroup. Similar to the results obtained for genetic structure, there were differences in the number and motifs of the members in EgrSnRK subfamily. Gene structure determines its function, and subfamilies of the *SnRK* gene family were involved in different plant growth stages and response to stresses.

In this study, 17 paralogous gene pairs were identified, two of which were involved in tandem duplication events (Table 2). Moreover, gene clusters were formed on chromosome 1, 2 and 7, respectively (Figure 5). Additionally, we identified 47 orthologous pairs between Eucalyptus and other species (i.e., *A. thaliana*, rice, poplar and *Vitis vinifera*). To further probe the selective pressure of *EgrSnRK* genes in evolution, we performed a sliding-window analysis of 17 paralogues gene pairs (Appendix A), indicating that different sites/encoding regions of gene pairs experienced differential selection pressures. The Ka/Ks ratios of the full-length coding sequences from 12 gene pairs were much lower than one, which indicated that most *EgrSnRK* genes underwent quite potent purifying selection. Meanwhile, this result showed that the sequence of the *EgrSnRK* gene family was highly conserved. In addition, we found that Ka/Ks in the domain regions of the encoded protein sequence was far lower than one, while Ka/Ks outside the domain regions was greater than one. Different protein domains in proteins are usually associated with different functions; thus, protein function and importance might exert a key influence on the rate of genetic evolution.

When plants are subjected to stress, through a series of signal transduction events that correspond to the appropriately stimulated transcription factors, those activated transcription factors bind to cis-acting elements of the target gene promoter, thereby activating the coordinated transcriptional expression of stress-resistant genes and creating regulatory responses to external stress signals [45,46]. Promoter analysis (Appendix A) showed that some *SnRK2s* and *SnRK3s* harbored GCN4_motif cis-regulatory elements that are involved in endosperm expression in their promoters. The result implied that SnRK2s and SnRK3s were likely to play key roles in development. Moreover, the earliest *SnRK1* of the plant was cloned from an endosperm cDNA library of the rye (Secale cereale L.) [14]. Moreover, some relatively recent studies had shown that SnRK2 and SnRK3 genes originated from the duplication of SnRK1 subfamily genes, and subsequently underwent rapid differentiation during evolution [15,24]. Most *SnRK* genes contain different types of cis-acting elements that are involved in plant growth, and various hormone and stress responses. Accumulating evidence suggests that *SnRK* genes are involved in the response to various abiotic stresses, including heat, drought and low temperatures. For example, *GhSnRK2* genes (11/20) from *G.hirsutum* responded to cold, heat, salt and drought by analyzing data freely available on a publicly available transcriptome database [22]. In *Hevea brasiliensis*, 7 of 10 *HbSnRK2s* had relative high levels of transcript abundance by ABA, ET, and JA [25]. Furthermore, eight *ZmCIPKs* were significantly up-regulated by at least one stress stimuli (salt, cold and heat) based on the public data of the maize found in the GEO database and semi-quantitative RT-PCR analysis [47].

To investigate the role of *SnRK2s* and *EgrSnRK3s* under salt treatment conditions, the expression levels of 31 *SnRK* genes were examined in the root, stem and leaf of *E. grandis* under various NaCl concentrations and time points under salt stress. Under conditions of a 200 mM salt treatment, it was found that 28 genes in the leaves, 20 in the root and 15 in the stem were up-regulated with increasing treatment time (Figure 8). It was worth noting that *EgrSnRK2.2*, *EgrSnRK2.4*, *EgrSnRK2.5*, *EgrSnRK3.2*, *EgrSnRK3.3*, *EgrSnRK3.5* and *EgrSnRK3.18* were strongly up-regulated in the leaves and root, but down-regulated in the stem. Moreover, Some *SnRK* genes peaked at 24 h or 168 h by salt treatment in different tissues, such as the relative expression level of *EgrSnRK3.3* at 24 and 168 h, which was five- and 10-fold that of the root, respectively. It was a similar observation to that that found for the *SnRK2* genes reported in cotton, wherein three genes were maximally at 24 h under salt treatment [23]. Both *EgrSnRK2.7* and *EgrSnRK2.8* were obviously (more than 3-fold) up-regulated after one hour and down-regulated at subsequent time points, where in the expression pattern was similar to that of *GmSnRK2.1* and *GmSnRk2.9* [22]. In the grapevine, the *VvSnRK2.13* and *VvSnRK2.7a* genes were strongly induced under conditions of salt stress; however, the expression of *VvSnRK2.14* was suppressed by exposure to salt stress [44]. Consistently, the *VvSnRK2.13* homolog *EgrSnRK2.1*was up-regulated more than five-fold at 1, 6, and 24 h, and more than 10-fold by 168 h. In addition, *EgrSnRK2.1* was up-regulated more than five-fold in different concentrations (Figure 7). *EgrSnRK2.6* is the orthologue of *VvSnRK2.14*, which was slightly induced under different time points and concentrations of salt. According to previous documents, most *SnRK2* and *SnRK3* genes from *A. thaliana*, rice, soybean, maize and distachyon, responded to salt treatment—an observation that indicated the important role that the SnRK family plays in response to stress and in enhancing abiotic stress tolerance of plants. Such as overexpression of *ZmCIPK21* can increase plant resistance to salt stress, and *BdSnRK2.9* improves tolerance to osmotic and salt stresses in transgenic tobacco [24,47]. The expression of *GhSnRK2.6* was shown to be up-regulated under salt conditions, and further demonstrated that *GhSnRK2.6* improved salt tolerance in transgenic upland cotton and *A. thaliana* [48]. In addition, TaSnRK2.9 positively regulates the response of transgenic tobacco plants to drought and salt stress [30]. The *EgrSnRK2.6* clusters with these gens, but was up-regulated to two-fold with increasing salt concentrations. Interestingly, the paralogous of *EgrSnRK2.6* and *EgrSnRK2.2* were strongly induced by 400 mM NaCl. Previous research showed that the *OsSAPK4* significantly improve salt tolerance in transgenic rice [26]. Moreover, in *A. thaliana TaSnRK2.4* enhanced salt tolerance [49]. As shown in Appendix A, *EgrSnRK2.3, EgrSnRK2.4, EgrSnRK2.8, TaSnRK2.4, OsSAPK4* and *ZmSnRK2.11* cluster together, while expression of *ZmSnRK2.11 was* negatively regulated following exposure to salt and drought stress. Both *EgrSnRK2.3* and *EgrSnRK2.4* were up-regulated by different concentrations of salt; however, *EgrSnRK2.8* expression was down-regulated under those conditions. Salt and many other abiotic stresses stimuli inhibits plant growth. Thus, plants have to dispose of these conditions by several mechanisms. One of the mechanisms is through SOS. Three *PtSOS2* genes were cloned and transformed into poplar to test its function [41]. Over-expression of *PtSOS2* improves salt tolerance by mediating osmotic protection and by inducing the antioxidant enzyme system. *EgrSnRK3.13, PtCIPK7* and *AtCIPK6* cluster together, and *AtCIPK6* is essential for plant growth and development under salt stress [34]. The expression of *EgrSnRK3.13* was significantly (more than 20-fold) up-regulated with increasing concentrations in the stem. In addition, *AtCIPK21* was reported to regulates osmotic and salt stress [37]. The *EgrSnRK3.9*, as a homologous pair of *AtCIPK21*, was up-regulated more than 10-fold at salt concentration of 200mM and 400mM. In summary, these results implied that *EgrSnRK2s* and *EgrSnRK3s* might play an essential role in the response to salt stress in *E. grandis*. This study provides available data for selecting available candidate genes in the *EgrSnRK* genes family and for further research on salt tolerance mechanism.

## 4. Materials and Methods

### 4.1. Sequence Retrieval and Gene Identification

Protein and cDNA sequences of 39 *SnRK* genes in *A. thaliana* were downloaded as determined when interrogating the Phytozome database (http://www.phytozome.net/). Rice protein and cDNA sequences were obtained from the Rice Annotation Project (RAP) (https://rapdb.dna.affrc.go.jp/). The whole *E. grandis*, grapevine and p. *trichocarpa* genome resources of the genome sequences, cDNA sequences and protein sequences, were downloaded from the Phytozome database. Local BLAST (E-value-5) searches were performed and based on the Hidden Markov Model (HMM) profile of SnRK gene domains from the Pfam database (http://pfam.janelia.org/search/sequence). All candidate sequences of *SnRK* genes were manually screened to leave only candidate genes containing the known conserved domains, which were further filtered based on the Pfam database [50], the NCBI Conserved Domain database (http://www.ncbi.nlm.nih.gov/Structure/cdd/wrpsb.cgi) [51] and the SMART database (http://smart.embl-heidelberg.de/) [52]. Bioinformatics analysis of *E. grandis* SnRK genes was performed and the number of amino acids, open reading frame (ORF) length, molecular weight (MW) and isoelectric point(pI) for each gene was obtained using ExPASy (http://www.expasy.ch/tools/pi_tool.html). The subcellular localizations of the *EgrSnRK* genes were predicted using WoLP PSORT (https://wolfpsort.hgc.jp/).

### 4.2. Multiple Alignment and Phylogenetic Analysis

The ClustalX 2.11 [53,54] software program was used for multiple sequence alignment of 34 SnRK full-length protein sequences from *E. grandis*. On the basis of alignment, a phylogenetic tree was constructed using the NJ method [53] in MEGA 7.0 and bootstrap analysis was performed using 1000 replicates for each node. An unrooted NJ tree of all SnRK protein sequences from *A.thaliana*, rice, grapevine, *p. trichocarpa*, maize and *E. grandis* was constructed using MEGA 7.0 [55].

### 4.3. Identification of Conserved Motifs and Analysis of Gene Structure

The online Gene Structure Display Server (GSDS: http://gsds.cbi.pku.edu.ch) [56] was performed with CDSs and their corresponding genomic DNA sequences to show the exon-intron organization of SnRK genes. To identify conserved motifs of EgrSnRK proteins, the Multiple Expectation Maximization for Motif Elicitation (MEME) online program [57] (http://meme.sdsc.edu/meme/itro.html) was used with the following parameters: number of repetition = any, maximum number of motifs = 20; and optimum motif length = 6–200 residues.

### 4.4. Chromosomal Location

The *E. grandis* chromosome size information and location information of the *SnRK* genes were obtained from the Phytozome database. The online Map Gene2 Chrom web v2 (http://mg2c.iask.in/mg2c_v2.0/) was implemented to map the chromosomal positions and relative distances of *EgrSnRK* genes.

### 4.5. Ka and Ks Analysis of Homologous Pair

The method for defining paralogues and orthologues according to the method of Blanc and Wolfe [58]. This approach was achieved by BLASTN analysis of all nucleotide sequences in each species [59]. In a species, paired sequences with more than 300 bp alignments and over 80 percent homology were defined as a pair of paralogues. To identify putative orthologues between two different species (A and B), each sequence from species A was searched against all sequences from species B using BLASTN. Meanwhile, each sequence from species B was searched against all sequences from species A. The two sequences were defined as orthologues whose reciprocal best hits were each within ≥300 bp of the two aligned sequences.

The synonymous (Ks) and non-synonymous (Ka) substitutions per site between duplicated genes pairs were calculated based on the previous research methods. The protein sequences of gene pairs were aligned by MEGA 7.0, and the results were subsequently used to calculate Ks and Ka substitution rates with DnaSP 5 software [60,61].

### 4.6. Cis-Elements in the Promoter Regions of EgrSnRK Genes

Upstream sequences (2Kb) of each EgrSnRK-coding sequence was downloaded from the Phytozome database. And then PlantCARE software permitted analysis of cis-element distributions (http://bioinformatics.psb.ugent.be/webtools/plantcare/html/) in promoter regions [62].

### 4.7. Plant Materials, Growth Conditions and Salt Treatments

*E. grandis* GL1 cloned plants were used to measure gene expression, which were cultured by hydroponics for six weeks. The seedlings were grown in the greenhouse under14/10 h cycles of light/dark conditions at 23/27 °C and a humidity of 70 percent (Research Institute of Tropical Forestry, Chinese Academy of Forestry, Guangzhou, China).

For salt treatments, the *E. grandis* clone GL1 seedlings were cultured with 200 mM NaCl solution instead of nutrient solution. All leaves, roots and stems were harvested at 0, 1, 6, 24 and 168 h after treatment, respectively. Furthermore, after 24 h of treatment with different concentrations of NaCl (0, 50, 100, 200, 400mM), the roots, stems and leaves were collected. All collected samples were wrapped in foil and immediately frozen in liquid nitrogen and stored at −80 °C for total RNA extraction.

### 4.8. RNA Extraction and qRT-PCR Analysis

According to the manufacturer’s instructions, total RNA was extracted from the leaves, roots and stems by using the Aidlab plant RNA kit (Aidlab Biotech, Beijing, China). The integrity and concentration of the RNA was detected by 1.5 percent agarose gel electrophoresis and NanoDrop™ One/OneC (ThermoFisher Scientific, USA). Further, the first strand of cDNA was synthesized using the Revert Aid First Strand cDNA Synthesis Kit (ThermoFisher Scientific, USA) based on the specification. Specific primers of the EgrSnRK genes were designed by Primer Premier 5.0 and then detected by NCBI (https://www.ncbi.nlm.nih.gov/tools/primer-blast/index.cgi?LINK_LOC = BlastHome). In addition, the *EgrEF1α* was used as a house-keeping control gene [63]. Sequence information of all primers is shown in the attached Appendix A. QRT-PCR was performed on the Applied Biosystems 7500 (ThermoFisher Scientific, USA) by using TB Green Premix Ex Taq II (Tli RNaseH Plus; TaKaRa Biotechnology) with a 20μL sample volume. The reaction system was as follows: 95 °C for 30 s, then 40 cycles of 95 °C for 5 s, and 60 °C for 34 s, followed by 95 °C for 15 s, 60 °C for 60s, and 95 °C for 15 s. For each sample, we conducted three biological and three technical replicates. The relative expression levels of each gene were calculated by the standard 2^−ΔΔCT^ [64] approach as compared with untreated control plants that were set as “1”. Processing the data and GraphPad 5 software was used to illustrate all of the data [65].

## 5. Conclusions

Accumulating documents suggest that the SnRK family plays an important role in plant growth and resistance. In this study, we presented a genome-wide identification of the EgrSnRK family in *Eucalyptus grandis*, including a phylogenetic tree, the gene structure, and the conserved motif, chromosomal location, and multiple sequence alignment. We identified 34 *EgrSnRK* genes and divided them into three distinct subgroups (i.e., EgrSnRK1, EgrSnRK2 and EgrSnRK3). Different subfamilies of the *SnRK* gene family had distinct conserved domains; however, all of the genes had a protein kinase domain at the N-terminal. Moreover, differential expression of *EgrSnRK* genes in the root, stem and leaf under different NaCl concentrations and duration of treatment were performed. From these studies, *EgrSnRK* genes showed various responses to salt treatments. In addition, these results laid a theoretical foundation for further study on the function of the *EgrSnRK* gene family and the differential tolerance of plants to salt.

## Figures and Tables

**Figure 1 ijms-20-02786-f001:**
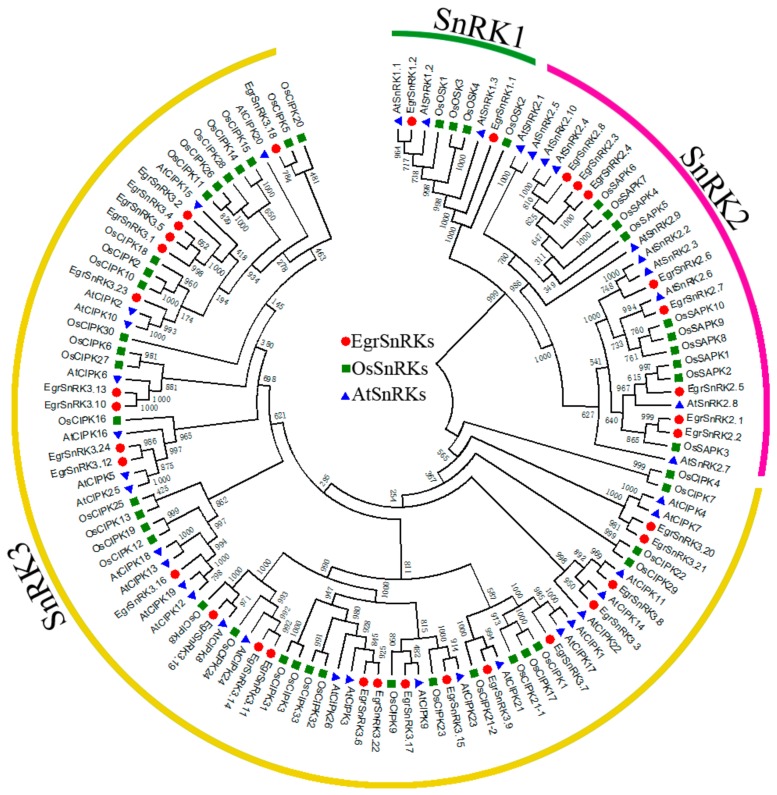
Phylogenetic tree of SnRK genes from Eucalyptus grandis, Arabidopsis, and rice. Thirty-four E*grSnRK* genes, 39 *AtSnRK* genes, and 48 *OsSnRK* genes are clustered into three subgroups (SnRK1, SnRK2 and SnRK3). The *SnRK* genes from *E. grandis*, *A. thaliana*, and rice are denoted by red, blue, and green, respectively. Details of the *SnRK* genes from all three plant species are listed in Appendix A. The tree was generated using Clustal X 2.0 software using the neighbor-joining (N-J) method.

**Figure 2 ijms-20-02786-f002:**
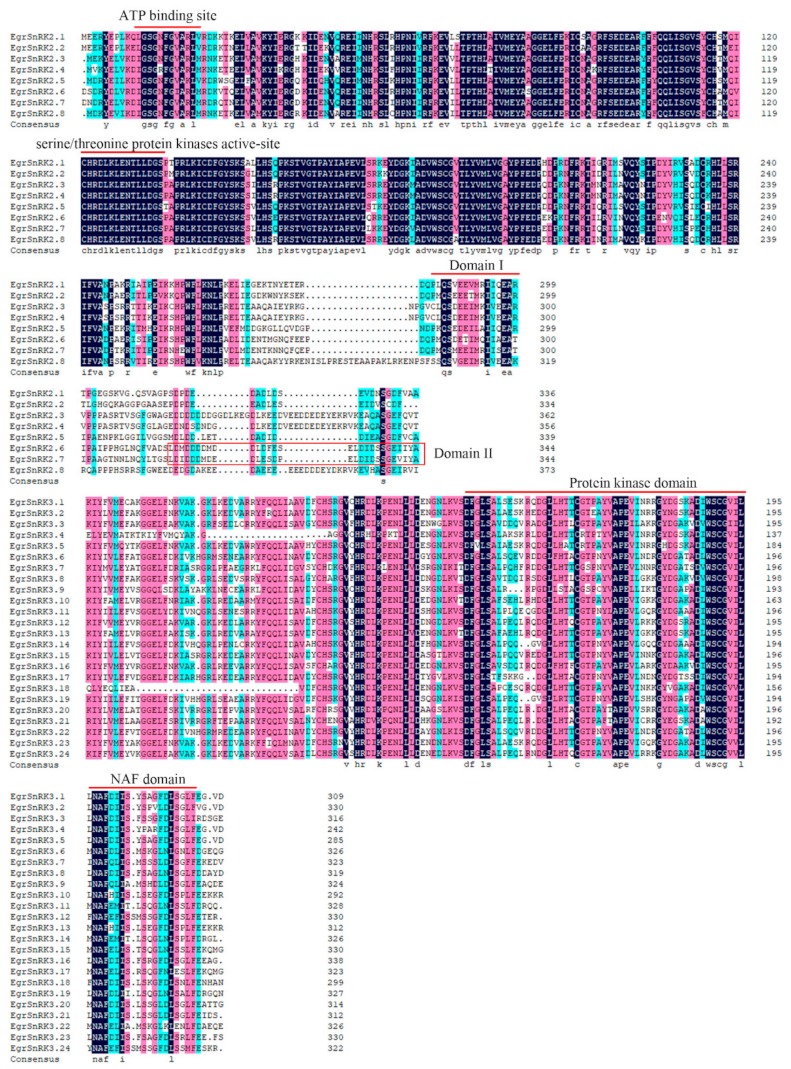
Multiple sequence alignment of *SnRK* genes in *E. grandis*. Sequences were aligned using DNAMAN 8 software.

**Figure 3 ijms-20-02786-f003:**
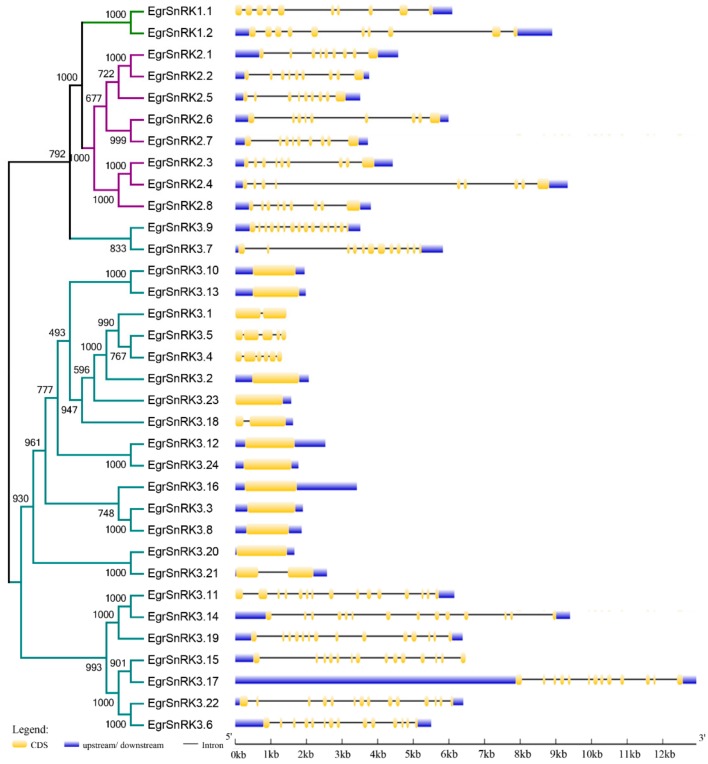
Phylogenetic relationships and gene structures of *SnRK* genes in *E. grandis*. *Left panel* shown the phylogenetic tree of *EgrSnRK* that was constructed by the neighbor-joining method. The SnRK1, SnRK2 and SnRK3 subfamily are marked by different colors. *Right panel* shown the gene structure of *EgrSnRK* genes. Exons are indicated by yellow rectangles. Gray lines connecting two exons represent introns.

**Figure 4 ijms-20-02786-f004:**
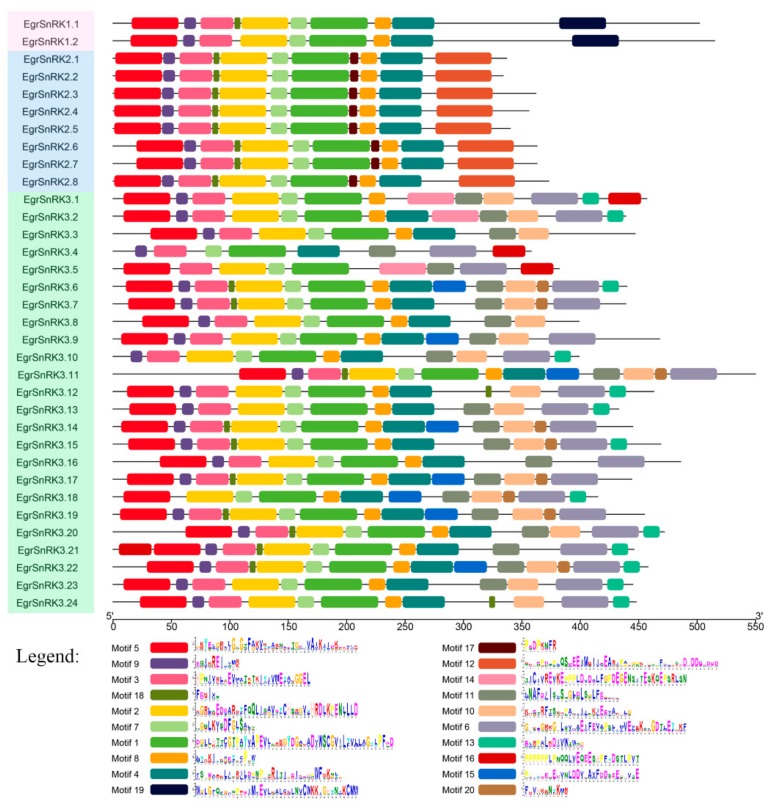
Conserved motifs of *SnRK* genes in *E. grandis*. Distribution of the 20 conserved motifs in the *EgrSnRK* genes following MEME analysis. The differentially colored boxes represent different motifs and their position in each sequence of *EgrSnRKs.*

**Figure 5 ijms-20-02786-f005:**
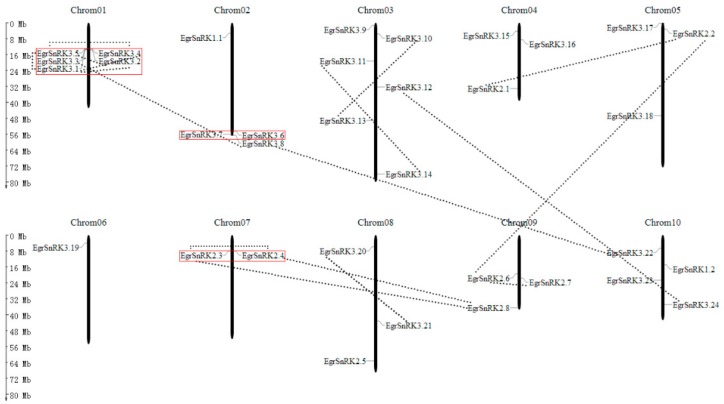
Chromosomal location of *EgrSnRK* genes. The 34 *EgrSnRK* genes are widely mapped to 10 of the 11 chromosomes. The paralogous pairs of *EgrSnRK* genes are connected by gray-dotted lines. The red boxes in front of the genes on behalf of these genes belonging to a gene cluster.

**Figure 6 ijms-20-02786-f006:**
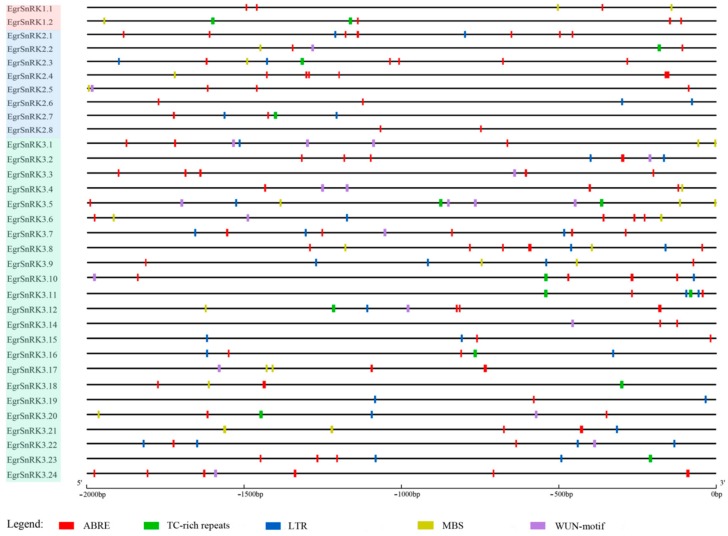
Cis-acting elements analysis of *EgrSnRK* genes in the promoter region. Gray lines represent promoter regions. Different color boxes represent different cis-acting elements. The ABRE elements are involved in the abscisic acid response, and TC-rich repeats elements are involved in defense and stress responses, the LTR elements are involved in low-temperature responses, the MBS elements are involved in drought-inducibility and the WUN-motif elements are involved in wound-responsive.

**Figure 7 ijms-20-02786-f007:**
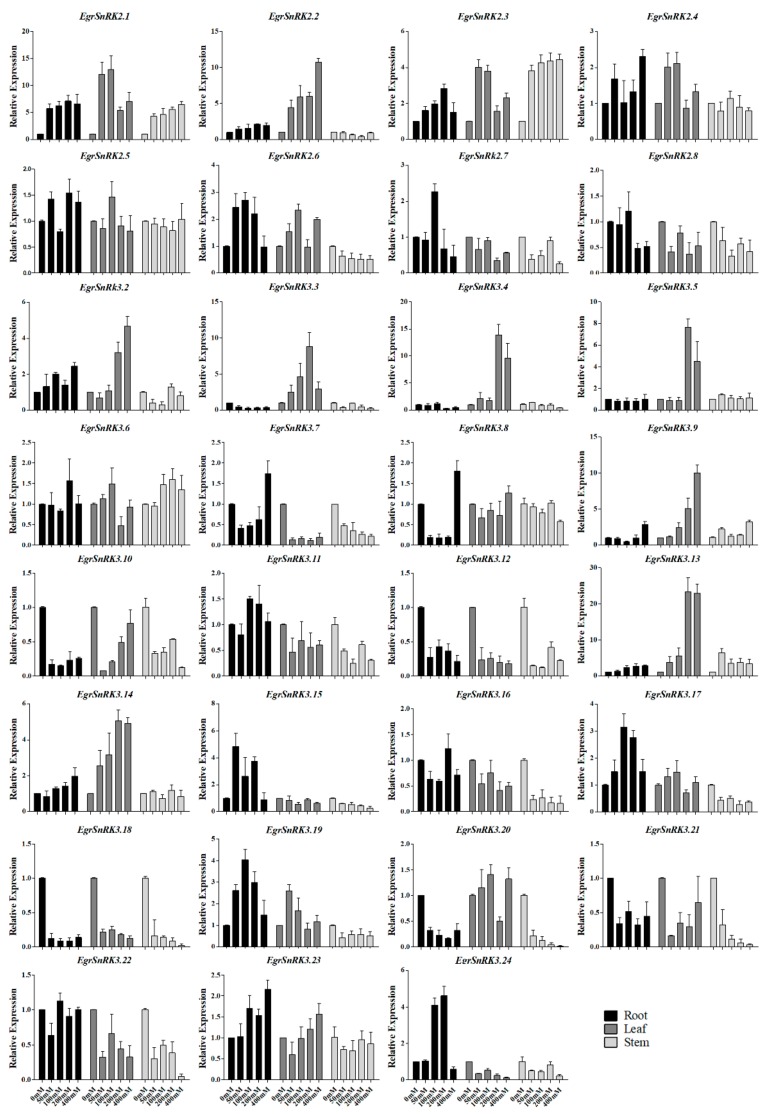
Expression analysis of 31 *EgrSnRK* genes following NaCl treatment at different concentrations as determined by qRT-PCR. The *Y*-axis and *X*-axis indicates relative expression levels and the time courses of stress treatments, respectively. Mean values and standard deviations (SDs) were obtained from three biological and three technical replicates. The error bars indicate standard deviation.

**Figure 8 ijms-20-02786-f008:**
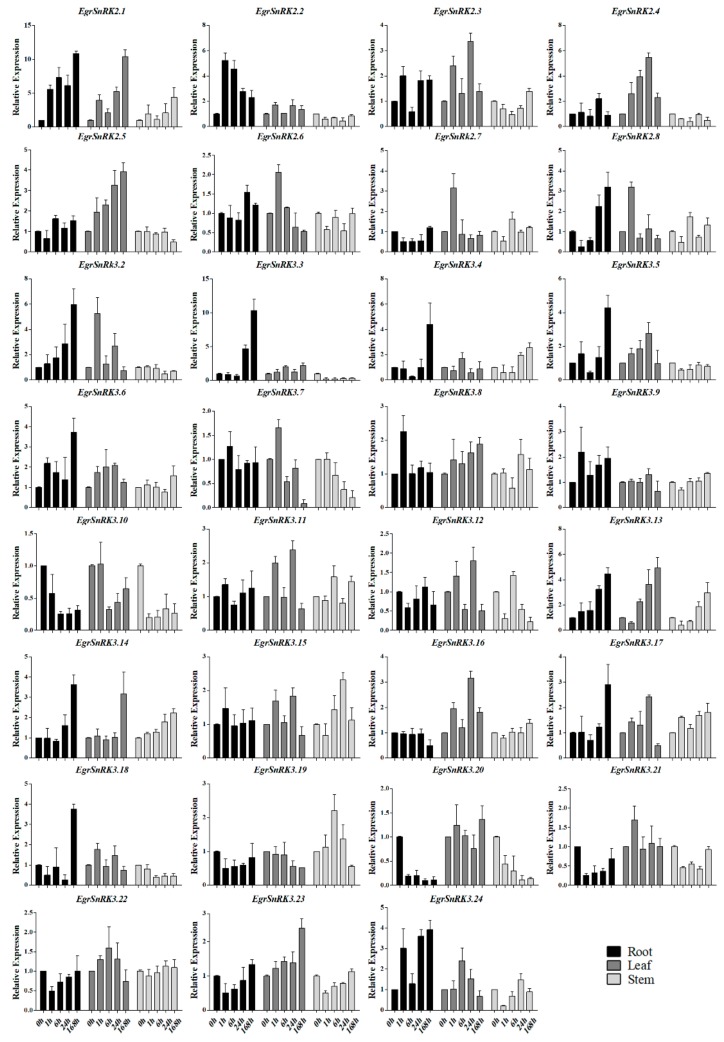
Expression analysis of 31 *EgrSnRK* genes following NaCl treatment at different time periods as determined by qRT-PCR. The *Y*-axis and *X*-axis indicates relative expression levels and the time courses of stress treatments, respectively. Mean values and standard deviations (SDs) were obtained from three biological and three technical replicates. The error bars indicate the standard deviation.

**Table 1 ijms-20-02786-t001:** List of SnRK genes that are identified in *Eucalyptus grandis*.

Name	Gene Identifier	Chr	Location Coordinates (5’–3’)	ORF Length (bp)	Protein			
					Length (a.a.)	PI	Mol.Wt. (kDa)	Exons
*EgrSnRK1.1*	Eucgr.B00544.1	2	5601699–5607791	1509	502	8.6	57.22	10
*EgrSnRK1.2*	Eucgr.J01364.3	10	15761839–15770736	1548	515	8.51	59.06	10
*EgrSnRK2.1*	Eucgr.D02135.1	4	34783964–34788536	1014	337	6.18	38.09	9
*EgrSnRK2.2*	Eucgr.E00345.1	5	3283351–3287110	1005	334	5.63	38.07	9
*EgrSnRK2.3*	Eucgr.G00557.1	7	8288459–8292881	1089	362	5.31	41.25	9
*EgrSnRK2.4*	Eucgr.G00558.1	7	8300485–8309815	1071	356	6	40.66	9
*EgrSnRK2.5*	Eucgr.H04745.1	8	66440405–66443914	1023	340	5.36	38.47	9
*EgrSnRK2.6*	Eucgr.I00977.1	9	20388568–20394557	1092	363	4.73	41.19	9
*EgrSnRK2.7*	Eucgr.I01180.1	9	22740915–22744636	1092	363	4.87	41.44	9
*EgrSnRK2.8*	Eucgr.I02742.1	9	38306416–38310220	1122	373	6.18	42.90	9
*EgrSnRK3.1*	Eucgr.A00690.1	1	14305972–14307407	1374	457	8.51	51.41	2
*EgrSnRK3.2*	Eucgr.A00691.1	1	14298350–14300413	1320	439	9.06	49.90	1
*EgrSnRK3.3*	Eucgr.A00711.1	1	14006453–14008352	1344	447	5.52	49.76	1
*EgrSnRK3.4*	Eucgr.A00734.1	1	13706777–13708090	1077	358	9.52	40.60	6
*EgrSnRK3.5*	Eucgr.A00737.1	1	13619140–13620568	1149	382	5.92	43.19	5
*EgrSnRK3.6*	Eucgr.B03773.1	2	57202252–57207754	1323	440	6.66	49.91	14
*EgrSnRK3.7*	Eucgr.B03958.1	2	5872963–58735460	1320	439	8.15	49.29	12
*EgrSnRK3.8*	Eucgr.B04021.1	2	59343262–59345126	1200	399	7	45.36	1
*EgrSnRK3.9*	Eucgr.C00193.1	3	1473208–1476720	1407	468	6	52.51	15
*EgrSnRK3.10*	Eucgr.C00357.1	3	5833911–5835857	1200	399	8.81	44.92	1
*EgrSnRK3.11*	Eucgr.C01333.1	3	20087289–20093442	1656	550	9.17	61.62	15
*EgrSnRK3.12*	Eucgr.C01944.1	3	33758197–33760724	1392	463	7.97	52.37	1
*EgrSnRK3.13*	Eucgr.C02590.1	3	51530920–51532898	1302	433	9.05	48.55	1
*EgrSnRK3.14*	Eucgr.C04226.1	3	79695224–79704623	1338	445	9.1	50.09	14
*EgrSnRK3.15*	Eucgr.D00272.1	4	4577009–4583478	1410	469	8.94	52.69	14
*EgrSnRK3.16*	Eucgr.D00481.1	4	8897498–8900910	1461	486	8.46	53.90	1
*EgrSnRK3.17*	Eucgr.E00017.1	5	222650–235591	1335	444	8.65	50.85	13
*EgrSnRK3.18*	Eucgr.E02758.1	5	48871914–48873537	1248	415	8.68	46.78	2
*EgrSnRK3.19*	Eucgr.F00453.1	6	4226550–4232934	1368	455	6.48	51.46	14
*EgrSnRK3.20*	Eucgr.H00223.1	8	6265751–6267413	1419	472	9.26	50.70	1
*EgrSnRK3.21*	Eucgr.H03182.1	8	45402293–45404868	1341	446	8.59	49.24	2
*EgrSnRK3.22*	Eucgr.J00641.1	10	7028075–7034476	1377	458	6.5	52.77	14
*EgrSnRK3.23*	Eucgr.J01840.1	10	23702496–23704068	1338	445	9.39	50.84	1
*EgrSnRK3.24*	Eucgr.J03116.1	10	36664597–36666371	1347	448	8.72	50.34	1

**Table 2 ijms-20-02786-t002:** Ka/Ks ratios of paralogous genes.

Paralogues		Ka (JC)	Ks (JC)	Ka/Ks
Gene 1	Gene 2			
*EgrSnRK2.2*	*EgrSnRK2.1*	0.0992	1.3197	0.075
	*EgrSnRK2.6*	0.23388	1.64249	0.142
*EgrSnRK2.3*	*EgrSnRK2.4*	0.03228	0.09785	0.33
	*EgrSnRK2.8*	0.17009	1.24632	0.136
*EgrSnRK2.4*	*EgrSnRK2.8*	0.13842	1.32215	0.105
*EgrSnRK2.6*	*EgrSnRK2.7*	0.11743	2.19481	0.054
*EgrSnRK3.1*	*EgrSnRK3.2*	0.11413	0.14826	0.77
	*EgrSnRK3.4*	0.54087	0.52272	1.035
*EgrSnRK3.5*	*EgrSnRK3.1*	0.08151	0.18191	0.448
	*EgrSnRK3.2*	0.06487	0.13926	0.466
	*EgrSnRK3.4*	0.15223	0.15972	0.953
*EgrSnRK3.3*	*EgrSnRK3.8*	0.0813	1.14857	0.071
*EgrSnRK3.6*	*EgrSnRK3.22*	0.35042	0.77327	0.453
*EgrSnRK3.10*	*EgrSnRK3.13*	0.10413	0.60606	0.172
*EgrSnRK3.11*	*EgrSnRK3.14*	0.13937	1.09741	0.127
*EgrSnRK3.12*	*EgrSnRK3.24*	0.19932	1.39897	0.142

**Table 3 ijms-20-02786-t003:** Numbers of each of the ABRE, TC-rich repeats, LTR, MBS and WUN-motif elements are shown.

Gene Name	ABRE	TC-Rich Repeats	LTR	MBS	WUN-Motif
*EgrSnRK1.1*	3			2	
*EgrSnRK1.2*	4	2		1	
*EgrSnRK2.1*	7		2		
*EgrSnRK2.2*	2	1		1	1
*EgrSnRK2.3*	5	1	2	1	
*EgrSnRK2.4*	10			1	
*EgrSnRK2.5*	3			1	1
*EgrSnRK2.6*	2		2		
*EgrSnRK2.7*	3	1	2		
*EgrSnRK2.8*	2				
*EgrSnRK3.1*	3		1	2	3
*EgrSnRK3.2*	5		2		1
*EgrSnRK3.3*	6				1
*EgrSnRK3.4*	4			1	3
*EgrSnRK3.5*	1	2	1	3	4
*EgrSnRK3.6*	6		1	2	1
*EgrSnRK3.7*	6		3		1
*EgrSnRK3.8*	5		2	2	
*EgrSnRK3.9*	2		3	2	
*EgrSnRK3.10*	6	1	1		1
*EgrSnRK3.11*	3	2	2		
*EgrSnRK3.12*	5	1	1	1	1
*EgrSnRK3.13*					
*EgrSnRK3.14*	2				1
*EgrSnRK3.15*	2		2		
*EgrSnRK3.16*	2	1	2		
*EgrSnRK3.17*	2			2	1
*EgrSnRK3.18*	2	1		1	
*EgrSnRK3.19*	1		2		
*EgrSnRK3.20*	3	1	1	1	1
*EgrSnRK3.21*	3		1	2	2
*EgrSnRK3.22*	3		5		2
*EgrSnRK3.23*	4	1	2		
*EgrSnRK3.24*	7				1

## Data Availability

The genome sequences of *E. grandis*, grapevine, *Populus trichocarpa*, and Arabidopsis were downloaded from Phytozome database (http://www.phytozome.net/). Rice protein sequences and cDNA sequences was provided by the Rice Annotation Project (RAP) (https://rapdb.dna.affrc.go.jp/).

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
