# Peer review of "Comprehensive Analysis of *SnRK* Gene Family and their Responses to Salt Stress in *Eucalyptus grandis"

_ijms, 2019, doi:10.3390/ijms20112786_

Round 1
Reviewer 1 Report
Wang et al. present a broad analysis of the SnRK genes in Eucalyptus grandis encompassing motif analysis, phylogenetic relationships and expression in response to salt treatment as inferred from RT-PCR. This work lays the foundation for future work on this family of genes in E. grandis with its key contributions including characterization of sequence motifs/domains and the qRT-PCR expression data for salt treatment at various time points and under different salt concentrations. Overall, I found the manuscript to be scientifically sound and clearly presented. Some questions and suggested edits are below:
1. Be sure to define all gene symbols/abbreviations. For example, CBL is first mentioned on line 43 and also mentioned later in the manuscript but was never defined.
2. Figure 4. Motifs that correspond to relevant protein domains (for example the NAF and kinase domains as discussed in lines 165-166).
3. Table 2. Currently, there are many comparisons and the relationships between the pairs are not immediately clear making it difficult to make interpretations. Table 2 could be grouped by paralog group to make it easier for the reader to interpret.
4. Figure 6. First, I think that it would be more appropriate to label the legend with the actual motif names instead of the predicted functions. Second, the criteria for considering a hit as ‘significant’ should be described. I have not used PlantCare before but it is surprising to me to see so many motifs detected. Just those displayed in figure 6 would have been plenty, but there are actually many more hits to other motif classes as listed in Table S6. Third, it would be easier for the reader to interpret figure 6 if a table was included beside the diagram indicating the counts of each of the motif classes.
5. Figure 7-8. Maybe I missed it, but why were the SnRK1s and SnRK3.1 not assayed?
6. Figure 9. I am not sure what the purpose of this figure is. What are we learning about the SnRK genes in E. grandis from the interactions of the Arabidopsis orthologs? It does not seem to be relevant to the work presented in this paper. I suggest removing this figure.
7. The authors state that SnRK2.1, 2.6, 3.9 and 3.13 are candidates for further studies on salt stress in E. grandis but I did not understand their rationale for this. Their expression patterns did not seem distinctly different from the other SnRK genes. The authors seem to be making their argument for these four genes as candidate genes in lines 330-372, but I found the text and reasoning here too long and convoluted to follow. I think this portion of the text needs to be condensed and rewritten to clearly explain why SnRK2.1, 2.6, 3.9 and 3.13 are more important than the other SnRK genes.
8. Lines 379-385. The authors seem to be saying that PFAM domains were analyzed with BLAST, but typically HMMER is used. Also, the authors state only that the PFAM, NCBI conserved domain and SMART databases were used but not how.
Author Response
Response to Reviewer 1 Comments
Point 1: Be sure to define all gene symbols/abbreviations. For example, CBL is first mentioned on line 43 and also mentioned later in the manuscript but was never defined.
Response 1: We sincerely thankful for the careful review from the reviewer. CBL is first mentioned on line 48 had been changed to calcineurin B-like (CBL) proteins.
Point 2: Figure 4. Motifs that correspond to relevant protein domains (for example the NAF and kinase domains as discussed in lines 165-166).
Response 2: I'm sorry we didn't explain it clearly. Each of the putative motifs obtained from MEME was annotated by searching Pfam. Motif 1, motif 2 motif 10, motif 11 were found to encode the kinase and NAF domains, while the other motifs have not function annotation. We add it in lines 179-182.
Point 3: Table 2. Currently, there are many comparisons and the relationships between the pairs are not immediately clear making it difficult to make interpretations. Table 2 could be grouped by paralog group to make it easier for the reader to interpret.
Response 3: Thanks a lot for this suggestion. According to the reviewer's suggestion, We have readjusted the Table 2.
Point 4:Figure 6. First, I think that it would be more appropriate to label the legend with the actual motif names instead of the predicted functions. Second, the criteria for considering a hit as ‘significant’ should be described. I have not used PlantCare before but it is surprising to me to see so many motifs detected. Just those displayed in figure 6 would have been plenty, but there are actually many more hits to other motif classes as listed in Table S6. Third, it would be easier for the reader to interpret figure 6 if a table was included beside the diagram indicating the counts of each of the motif classes.
Response 4: We are sincerely thankful for the comments. According to the reviewer's suggestion, we modified Figure 6 and added a description to the legend (lines 231-237 ). And we described ‘significant’ in revised manuscript.The counts of each of the motif classes in Table3.
Point 5: Figure 7-8. Maybe I missed it, but why were the SnRK1s and SnRK3.1 not assayed?
Response 5: We are sincerely thankful for the comments. SnRK1 is homologous to yeast and mammals, which plays an important role in regulating carbon metabolism and energy status. SnRK2 and SnRK3 subfamilies are unique to plants, which response to different abiotic stress. So we didn't choose the SnRK1 subfamily. The expression of EgrSnRK3.1 was not detected in this experiment.
Point 6: Figure 9. I am not sure what the purpose of this figure is. What are we learning about the SnRK genes in E. grandis from the interactions of the Arabidopsis orthologs? It does not seem to be relevant to the work presented in this paper. I suggest removing this figure.
Response 6: According to the reviewer's suggestion, we delete this part.
Point 7:The authors state that SnRK2.1, 2.6, 3.9 and 3.13 are candidates for further studies on salt stress in E. grandis but I did not understand their rationale for this. Their expression patterns did not seem distinctly different from the other SnRK genes. The authors seem to be making their argument for these four genes as candidate genes in lines 330-372, but I found the text and reasoning here too long and convoluted to follow. I think this portion of the text needs to be condensed and rewritten to clearly explain why SnRK2.1, 2.6, 3.9 and 3.13 are more important than the other SnRK genes.
Response 7: Thanks a lot for this suggestion. We combined phylogenetic analysis and qPCR data with homologous genes that have been identified for functional expression, and eventually we chose these four genes for our next work. According to the reviewer's suggestion, the next work plan should not be mentioned here. We decided to delete this part.
Point 8: Lines 379-385. The authors seem to be saying that PFAM domains were analyzed with BLAST, but typically HMMER is used. Also, the authors state only that the PFAM, NCBI conserved domain and SMART databases were used but not how.
Response 8: We are sincerely thankful for the comments. All candidate protein sequences of SnRK genes were uploaded to PFAM, NCBI conserved domain and SMART databases. Then we can get results and manual screened to leave only candidate genes containing the known conserved domains. HMMER is based on a Hidden Markov Model for biological sequence alignment, similar to BLAST. We get the HMM from PFAM and blast it locally. Our reference is as follows:
1. Danmei Chen, Zhu Chen, Min Wu et al., J Plant Growth Regul, Genome-Wide Identification and Expression Analysis of the HD-Zip Gene Family in Moso Bamboo (Phyllostachys edulis). 2017, 36, 323-337.

Reviewer 2 Report
The authors characterized the SNRK2 gene family from E. grandis including a bioinformatic characterization associated with gene expression study by RT-qPCR following NaCl treatment in 3 different tissue.
This family is already well charcaterized and no real novelty, except the characterization of this gene family in this new species are here provided. However, we also have to consider that few studies deal with trees as compare to crops, this point is in favor of the present study. However the authors did not emphazised this major point in their study. I suggest to the authors to highlight this point.
The other point is: I didn't understand why the authors porposethis work as a "communication" and not as a "regular paper"?
Here are some comments on the manuscript.:
L10: The not in bold
L44: calcineurin B-like protein-interacting protein kinases, not « balcineurin B-like protein-interacting protein kinases »
Paragraph 2.1: usually molecular weights are provided in kDa not in Da.
L146: delete “And”
L169: motifs 12 and 13 are specific of SNRK2 family, not motifs 12 and 19 as written. Motif 19 is specific of SNRK1 family.
Table 2: please calculate duplication time
RT-qPCR analysis: in my opinion it would be better to present the results of the different NaCl concentrations before the time course measurement.
L315: add space before “Promoter”
L346: B. distachyon already cited so no need to write Brachypodium.
L473: Populus trichocarpa, not « populous trichocarpa »
L487: Populus trichocarpa not « populus trichocarpa »
Author Response
Response to Reviewer 2 Comments
Point 1: This family is already well charcaterized and no real novelty, except the characterization of this gene family in this new species are here provided. However, we also have to consider that few studies deal with trees as compare to crops, this point is in favor of the present study. However the authors did not emphazised this major point in their study. I suggest to the authors to highlight this point. The other point is: I didn't understand why the authors porposethis work as a "communication" and not as a "regular paper"?
Response 1: We are sincerely thankful for the enlightening comments. The type of article I choose is Article, is not communication.
Point 2:L10: The not in bold
Response 2: We sincerely thankful for the careful review from the reviewer. We have modified the"The"to "The" in line 11.
Point 3: L44: calcineurin B-like protein-interacting protein kinases, not « balcineurin B-like protein-interacting protein kinases »
Response 3: We have changed "balcineurin B-like protein-interacting protein kinases" to "calcineurin B-like protein-interacting protein kinases" in revised manuscript (L49).
Point 4: Paragraph 2.1: usually molecular weights are provided in kDa not in Da.
Response 4: Many thanks for this comment. We changed all the units of molecular weight in the manuscript to kDa.
Point 5:L146: delete “And”
Response 5: Many thanks for this comment. We delete “And” (L150).
Point 6:L169: motifs 12 and 13 are specific of SNRK2 family, not motifs 12 and 19 as written. Motif 19 is specific of SNRK1 family.
Response 6: Thanks a lot for this suggestion. Due to our carelessness, we made a mistake. We amend the sentence in revised manuscript (L185).
Point 7: Table 2: please calculate duplication time.
Response 7: Many thanks for this comment. The Ks value was translated into divergence
time (T) in millions of years using the formula T = Ks/2λ*10-6 Mya. But we consulted many literature and did not get the λ of Eucalyptus grandis.
Point 8: RT-qPCR analysis: in my opinion it would be better to present the results of the different NaCl concentrations before the time course measurement.
Response 8: According to the reviewer's suggestion, we present the results of the different NaCl concentrations before the time course measurement.
Point 9:L315: add space before “Promoter”
Response 9: We sincerely thankful for the careful review from the reviewer. We add space before “Promoter”(L332).
Point 10:L346: B. distachyon already cited so no need to write Brachypodium.
Response 10: "Brachypodium" was corrected as "distachyon" (L367).
Point 11:L473: Populus trichocarpa, not « populous trichocarpa »
Response 11: We have changed "populous trichocarpa" to "Populus trichocarpa" in revised manuscript (L400).
Point 12:L487: Populus trichocarpa not « populus trichocarpa »
Response 12: We have changed "populous trichocarpa" to "Populus trichocarpa" in revised manuscript (L509).

Reviewer 3 Report
Manuscript by Wang et al. is poorly written right from the Abstract to the discussion. Authors even did not bother to look for the spelling mistakes throughout the manuscript. The entire article is not appealing to read and understand due to poor grammar.
However, I appreciate the experimental design and extensive analysis to profile SnRKs in E. grandis, but the quality of presentation does not reach to the impact of international journals.
I advise authors to improve the quality of writing and presentation to make it readable. One more advice for the analysis part is to generate a phylogenetic tree with E. grandis with comparison to Arabidopsis or rice SnRKs extensively studied to simplify the tree.
Author Response
Response to Reviewer 3 Comments
Point 1: Manuscript by Wang et al. is poorly written right from the Abstract to the discussion. Authors even did not bother to look for the spelling mistakes throughout the manuscript. The entire article is not appealing to read and understand due to poor grammar.
However, I appreciate the experimental design and extensive analysis to profile SnRKs in E. grandis, but the quality of presentation does not reach to the impact of international journals.
I advise authors to improve the quality of writing and presentation to make it readable. One more advice for the analysis part is to generate a phylogenetic tree with E. grandis with comparison to Arabidopsis or rice SnRKs extensively studied to simplify the tree.
Response 1: we have the language polished by the native English scientist. We want to compare Eucalyptus with Arabidopsis and rice. And we also want to compare Eucalyptus with poplar, which are related to each other.

Reviewer 4 Report
Review Comments The objectives and background of this manuscript are clear. The manuscript is well written. However, the following revisions should be carried out; - Abstract Other important analyses carried out in this manuscript should be included in the abstract. - Introduction The introduction is clear, but not covered the story well. The introduction should cover the literature on this topic as well. Also, recent references should be included. - Results The results are well explained. However, Figure 3 should be explained in details. The authors should also indicate the significant findings arisen from Figure 4. - Discussion The discussion is somewhat well-written but still needs improvement in sections 3 and 4. The current data should be compared with previously published findings and how these new findings support the research question. - Methods Different sections should include more details on how you made such analysis, i.e. software, …etc -References Up to date references should be included to reveal the up to date information that could support these findings as well.Author Response
Response to Reviewer 4 Comments
Point 1: The objectives and background of this manuscript are clear. The manuscript is well written. However, the following revisions should be carried out; - Abstract Other important analyses carried out in this manuscript should be included in the abstract. - Introduction The introduction is clear, but not covered the story well. The introduction should cover the literature on this topic as well. Also, recent references should be included. - Results The results are well explained. However, Figure 3 should be explained in details. The authors should also indicate the significant findings arisen from Figure 4. - Discussion The discussion is somewhat well-written but still needs improvement in sections 3 and 4. The current data should be compared with previously published findings and how these new findings support the research question. - Methods Different sections should include more details on how you made such analysis, i.e. software, …etc -References Up to date references should be included to reveal the up to date information that could support these findings as well.
Response 1: We are sincerely thankful for the comments. We have asked help from a native English scientist to polish the language of this revision. We have carefully revised our paper according to suggestions provided by the reviewers.

Round 2
Reviewer 3 Report
The revised manuscript by Wang et al. is substantially improved. English editing indeed increased the quality of the presentation. Minor tweaks in sentence structure further enhanced the scientific soundness.
I am still not convinced by authors of generating EgSnRKs phylogenetic tree with three other crop species. I feel like the tree will be more crowded with genes ID. Moreover, including SnRKs from more crop species to compare with EgSnRKs will not lead to any significant conclusion. One option is that authors can separately generate a tree for EgSnrks in comparison with poplar SnRks as a new figure to enhance the clarity of the tree. However, I leave the final decision for authors, and this my comment will not affect the final decision of the manuscript.
Author Response
Response to Reviewer 3 Comments
Point 1: I am still not convinced by authors of generating EgSnRKs phylogenetic tree with three other crop species. I feel like the tree will be more crowded with genes ID. Moreover, including SnRKs from more crop species to compare with EgSnRKs will not lead to any significant conclusion. One option is that authors can separately generate a tree for EgSnrks in comparison with poplar SnRks as a new figure to enhance the clarity of the tree. However, I leave the final decision for authors, and this my comment will not affect the final decision of the manuscript.
Response 1: We are sincerely thankful for the comments. According to the reviewer's suggestion, we have readjusted the Figure 1. In addition, phylogenetic tree of SnRK genes from Eucalyptus grandis, Arabidopsis, rice, grapevine and populous trichocarpa was illustrated in Figure S1.
